

# Diagnostic structure of visual robotic inundated systems with fuzzy clustering membership correlation

Hariprasath Manoharan[1], Shitharth Selvarajan[2], Rajanikanth Aluvalu[3], Maha Abdelhaq[4], Raed Alsaqour[5] and Mueen Uddin[6]

[1] Department of Electronics and Communication Engineering, Panimalar Engineering College, Poonamallee, Chennai, Tamil Nadu, India
[2] Department of Computer Science, Kebri Dehar University, Ethiopia
[3] Department of IT, Chaitanya Bharathi Institute of Technology, Hyderabad, India
[4] Department of Information Technology, College of Computer and Information Sciences, Princess Nourah bint Abdulrahman University, Riyadh, Saudi Arabia
[5] Department of Information Technology, College of Computing and Informatics, Saudi Electronic University, Riyadh, Saudi Arabia
[6] College of Computing and IT, University of Doha for Science and Technology, Qatar

Corresponding author
Rajanikanth Aluvalu,
rajanikanth.aluvalu@gmail.com

## ABSTRACT

The process of using robotic technology to examine underwater systems is still a difficult undertaking because the majority of automated activities lack network connectivity. Therefore, the suggested approach finds the main hole in undersea systems and fills it using robotic automation. In the predicted model, an analytical framework is created to operate the robot within predetermined areas while maximizing communication ranges. Additionally, a clustering algorithm with a fuzzy membership function is implemented, allowing the robots to advance in accordance with predefined clusters and arrive at their starting place within a predetermined amount of time. A cluster node is connected in each clustered region and provides the central control center with the necessary data. The weights are evenly distributed, and the designed robotic system is installed to prevent an uncontrolled operational state. Five different scenarios are used to test and validate the created model, and in each case, the proposed method is found to be superior to the current methodology in terms of range, energy, density, time periods, and total metrics of operation.

## INTRODUCTION

The underwater communication systems, which are used to monitor all types of objects, obstructions, water levels, and the healthiness conditions of aquatic organisms, process the majority of the advanced technologies that enable wireless data transfer operations. However, the majority of cutting-edge technologies only allow wireless devices to be connected without an exterior covering, making it impossible for the complete communication system to establish accurate observations. Even when an exterior shell is offered, some networks simply use unrestrained methods to function. In order to enable the observation processes employing a robotically connected automatic data transfer network, more sophisticated technology must be developed. As a result, the suggested

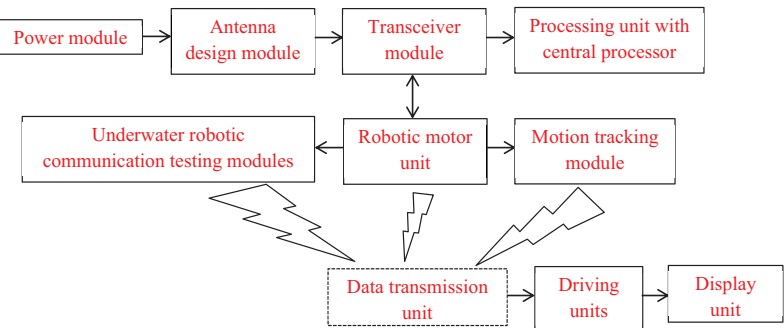

**Figure 1  Block diagram of underwater robotic units.**

method creates a robotic system based on cutting-edge underwater architectures where the whole set of underwater resource characteristics may be examined with a variety of potential uses. The motion level is unaffected by any external characteristics when using the projected technique because moving robots receive no input in the form of audio or video samples. However, as the robots can only move in a circular pattern, a control center is used to operate all of the deployed robots so that the movements made by each one can be observed. The entire region will be covered as a result of proper movement detection, making the categorization technique more applicable for large-scale spaces. Even though heavy work-class operations are always guaranteed, there is no guarantee that pre-training approaches will be included.

Additionally, all robotic actions depend on sending information regarding involvement duties and other defined functionality to a central server. The proposed working principle's block diagram is shown in Fig. 1. The connected representations of Fig. 1 show an antenna interface unit coupled to a power supply unit with both primary and secondary backups. As a result, an antenna transceiver unit that processes the operation using a central CPU and wireless transmissions is used to operate the entire robotic system. When tracking and testing are finished, the transceiver module is coupled with the motor units to connect the initial portion of transmission. A data transmission unit is attached to provide the data to the robotic driving units once the robotic motions have been monitored and the associated segments have been reported to the control center. The data from the robotic system is connected to the output control unit at the end of the observation process in order to view the parametric results.

## BACKGROUND AND RELATED WORKS

Many IoT researchers have created various gadgets for analyzing underwater systems where wireless media is used for data transmission. As a result, this section offers an overview of all recent publications that offer primary data about undersea systems using various methodologies. The majority of solutions make use of sophisticated optimization algorithms to assist underwater systems that have been created. The proposed strategy for creating the analytical framework and selecting advanced models rather than current systems will be supported by the literature review. In *Kong et al. (2020)*, a hybrid model is created to support underwater systems using solar cell technology without the need for

high power for state monitoring. However, the underwater solar cells must be properly covered with a substance to prevent excess power generation. Even if the hybrid model has been modified and installed as a self-contained monitoring system, the lack of an intelligent detection system prevents the model from taking the proper measurements. A probabilistic architecture is therefore introduced with hydrophone sensor networks for appropriate measurements (*Kshirsagar et al., 2022*). All pertinent data can be easily gathered because the probabilistic model offers a relative set of measurements about various objects that are present deep under the sea. A special underwater vehicle is designed and shown in addition to deep sea monitoring operations, although the issue of floating for vehicle installation has not been addressed. Therefore, a robotic model for a spherical underwater system is created to prevent fluctuations in the deep sea (*Yue, Guo & Shi, 2013*). All hydrodynamic features and data will be tracked if a better propulsion system is installed, allowing for rapid jet propulsion to occur. However, without using any data handling techniques, the hydrodynamic features can be observed in the depths of the ocean.

In order to prevent various disturbances in wireless systems, a data handling strategy has been developed with a state-independent evaluation process (*Rostami et al., 2018*). Furthermore, the state-dependent technique is used to prevent real-time intrusion found in underwater systems, allowing for easy identification of all items. Energy restrictions are also an issue since obstacle avoidance still requires high power transfer underwater. Solar cells are therefore used in underwater monitoring systems to account for the energy constraint even in different water settings (*Enaganti et al., 2020*). However, some solar cells may drift beneath the sea, and it's possible that connections may be cut off without any reference points, leading to a challenging situation during the monitoring stage. A distinct node redeployment technique is developed and integrated with the system model to prevent the complex issues that are prevalent in monitoring operations (*Jiang, Feng & Wu, 2016*). Although the redeployment strategy offers some insight pertaining to coverage values, it is unable to offer data on the density of each node that is connected to a power source. Even at low energy levels, it is feasible to avoid interference from various obstacles, and this needs to be stated in the underwater network architecture. A greater number of co-efficient are enabled with various scattering matrix representations in order to provide direct specification values (*Woźniak, Darecki & Sagan, 2019*). Due to the complicated water that makes up the underwater monitoring system, a unique empirical link with sea scattering formulas has been created. By allowing the installation of the pre-defined data set, the complex operation can be transformed into an application-dependent operation as a result of the representation of such variables. Even if conversions have been done and the design process is considerably simpler, a specific set of rule procedures must be used to reflect some of the pre-defined assumptions.

It is feasible to offer a simulation module that is based on throughput representations by removing the pre-defined assumption (*Salawu, Bright & Onunka, 2020*). In this form of design, the entire robotic system's behaviour is tested, and the entire manufacturing facility

**Table 1 Comparison of existing literature.**

| References | Active methods/algorithms | Objectives | | | | |
|---|---|---|---|---|---|---|
| | | A | B | C | D | E |
| *Sosa et al. (2020)* | Parasitic resistance framework for underwater systems | ✓ | | ✓ | | |
| *Wang et al. (2021)* | Artificial intelligence algorithm | | ✓ | ✓ | | |
| *Shetty, Pai & Pai (2018)* | Software as a service with resource planning | ✓ | | | ✓ | |
| *Sandøy & Schjølberg (2017)* | Extended Kalman filter for underwater detection | | ✓ | | | ✓ |
| *Jorge et al. (2021)* | Sampling approach framework | | ✓ | | ✓ | ✓ |
| *Gilbert (2021)* | Slender robotic configuration | ✓ | | ✓ | | ✓ |
| *Singh et al. (2015)* | Biometric thruster robot | ✓ | ✓ | ✓ | | |
| *Diao & Wei (2022)* | Multiple data representations with fuzzy environments | | ✓ | | ✓ | |
| *Zhang et al. (2022)* | Kinematic problem solving for all environments using robotic movements | ✓ | | ✓ | ✓ | |
| *Chen et al. (2022)* | Inspection robot design for closed cabinets with fuzzy membership functions | | ✓ | | | ✓ |
| *Christensen et al. (2022)* | Control of underwater using artificial intelligence algorithms | | ✓ | | ✓ | ✓ |
| Proposed | Underwater robotic system with analytical framework | ✓ | ✓ | ✓ | ✓ | ✓ |

**Note:**
A, communication ranges; B, node density; C, energy maximization; D, monitoring time periods; E, maximization of individual scores.

is modified. As a result of these adjustments, the robot can quickly take and position various resources that are present in underwater systems. The classical representation model is used to assume all parametric values at this robotic testing stage, saving the entire production time. The researchers (*Câmara Júnior, Vieira & Vieira, 2020*) have developed a data gathering algorithm that is used to deliver complete data to sink nodes in order to improve the robotic operation. With such algorithms, only acoustic networks can be formed, and the various latencies present at each node can be reduced by roughly 73 percent by using the technique for data collection. Since a non-intrusive form of representation will be left empty, the entire underwater operation can be modified by employing two modes with clever sensor detection processes (*Chang et al., 2022*). The majority of imaging devices won't function well in an empty space; hence, sonar representation systems should be used instead of wireless modules. The data modules are very complex, even when sonars are employed, and failure will happen when nodes alter. In order to conduct an independent investigation of energy use, the entire network was modified to operate with low robustness (*Sosa et al., 2020*). However, distinct parameters must be defined, resulting in low operating states throughout the entire operation. Additionally, recent research with objective functions is given in Table 1.

## RESEARCH GAP AND MOTIVATION

There are any identical existing that considers robotic system as major platform for various applications where data processing system is connected for providing uninterrupted data to receiver. However the major gap that is identified in existing approaches is that the robotic systems even if considered for underwater monitoring cannot able to establish a

reliable data connection with control centre. Due to above mentioned drawback the information about treacherous objects or other form of spices that is present under the sea is not transmitted to central units. Hence a significant solution must be provided to all the gap that is present in consideration of robotic systems.

- Whether the robotic system can be built with additional communication ranges by installing nodes are various points at a distance below water levels.
- Can the system measurements with robots be increased with reduction of inappropriate values? Thereby performance of robotic systems can be maximized.
- Is the designed system able to operate the robotic systems for underwater operations with minimized energy rates even if densities across certain ranges are maximum?

## Novelty

The entire design of robotic system in proposed method is made using a novel mathematical model which is arranged into separate clusters. For each cluster (fuzzy membership function) the data is transmitted to control center which describes the current state operation of underwater systems. Once the data is transmitted by several clusters then hierarchical procedure is followed for arranging entire data segments in proper format thereby ensuring appropriate metrics for each transmitted data. Moreover the major novelty of projected model is to maximize the detection range of robots that is present at underwater systems. In addition as compared to existing models applicable point of detection is made by robotic systems thus time period of monitoring is reduced. Further the relationship in terms of analytical parameter which is made with respect to distinct subsurface area provides precise fuzzy membership function. The above mentioned fuzzy membership function in proposed method is calculated by considering the functions for first and last node points.

## Major contributions

To provide solutions for aforementioned research gap in the field of robotics for underwater communication the proposed method is incorporated by using a cluster node at various points deep inside the concentration levels where every data at control centre is transmitted by using fuzzy logic controller that provides the advantage of clustering in underwater communications. The novelty of the projected model depends on measuring all parameters that can be either minimized or maximized.

- To design the robotic system that can able to provide maximized communication ranges for data transmission after identifying corresponding targets.
- To maximize the performance of robotic systems where improbable values are reduced by measuring accurate points (in terms of identification and node implementations).
- To reduce the overlap conditions of installed nodes by using fuzzy clustering algorithm thereafter maximizing the energy ranges at each node points to provide uninterrupted data about underwater systems.

### Article organization

The remaining section of the article is organized as follows, "Proposed system model" provides detailed mathematical model for proposed robotic system. "Methods/Experimental" integrates the designed robotic system with fuzzy membership function and data path measurements are made using clustering phases. "Results and discussion" simulates the outcome of proposed method with comparison cases. Finally the conclusion is provided with limitations and directions for future work.

### Proposed system model

A special framework design is required for underwater systems in order to reduce operational complexity. Therefore, the operational conditions of robotic systems using sensor modules are provided along with an IoT framework in this part. To avoid errors that occur during the installation's initial phase, measurements are conducted precisely when defining the analytical terms. Furthermore, if a robotic model is represented, specific parameters derived from Eq. (1) must be used to define the robot's communication range.

$$C_i = max \sum_{i=1}^{n}(N_i - T_i) \tag{1}$$

where,

$N_i$ denotes the installed node module.

$T_i$ indicates target in underwater systems.

The difference between the installed node and the matching target, which needs to be maximized over a specific distance, is represented by Eq. (1). Since Eq. (2) can be used to measure the performance of robotic tools, some positions must be tested using this range.

$$P_i = max \sum_{i=1}^{n} I_i * (U_i - U_1) \tag{2}$$

where,

$I_i$ describes the input parametric values.

$U_i$, $U_1$ denotes improbability values of robotic measurements.

The second objective function, which must be maximized using an input parameter and a set of unknown measurements, is represented by Eq. (2). Equation (3) can be used to define a node parameter as follows to prevent uncertain measurements,

$$MN_i = \sum_{i=1}^{n} \frac{TN_i}{RN_i} * 100 \tag{3}$$

where,

$TN_i$ indicates total number of nodes installed in underwater.

$RN_i$ denotes minimum number of node coverage in defined radius.

Equation (3) calculates the percentage of the system's total uncertain periods that are avoided when compared to uncertain measurement data. However, if the installed nodes

overlap, the concentration in the robotic network will be low and needs to be optimized using Eq. (4), as stated below,

$$density_i = max \sum_{i=1}^{n} \frac{V_i}{P_i} \tag{4}$$

where,

$V_i$ indicates the volume of node.

$P_i$ denotes individual node partition.

The concentration range of the robot will expand in underwater systems as additional nodes are divided, resulting in the proper energy consumption, which is described using Eq. (5) as follows,

$$Energy_i = max \sum_{i=1}^{n} \tau_i * d_t(i) * \rho_i \tag{5}$$

where,

$\tau_i$ denotes supplied power for robotic operation.

$d_t$ indicates time required by a robot for data transmission.

$\rho_i$ represents robotic network transmission distance.

Equation (5) depicts the period of operation at maximum efficiency, during which the issue of more power causes an increase in the likeness ratio, which can be characterized using Eq. (6) as follows,

$$L_i = \sum_{i=1}^{n} \frac{SS_b(i)}{SS_a(i)} * 100 \tag{6}$$

where,

$SS_b$, $SS_a$ denotes subsurface area that is present at both levels in underwater systems.

Equation (7) can be used to express the overall time required to provide acceptable service using the complete network and nodes. The likeness ratio as mentioned in Eq. (6) must be normalized with regard to the motions of the robot.

$$Time_i = \sum_{i=1}^{n} m_t(i) + \beta_t(i) + \varphi_i \tag{7}$$

where,

$m_t$ indicates mean time period.

$\beta_t$ represents time taken for robot envisioning from underwater.

$\varphi_i$ denotes cyclic time periods.

Using signaling systems with an inter-arrival rate, the whole time of the robotic system may be observed, resulting in a significantly shorter representation time than would be anticipated. Equation (8) can be used to express the objective function by merging the full analytic framework.

$$obj_i = max \sum_{i=1}^{n} C_i, P_i, density_i, Energy_i \tag{8}$$

For expanding the monitoring communication range, the density of monitoring in subsea systems, the applied power for monitoring, and the energy of node systems, the objective function defined in Eq. (8) is a maximization framework. However, an integration segment with an optimization method is necessary for underwater testing and is explained in "Research gap and motivation" in order to maximize the effectiveness of the suggested technique. In the proposed method there are four different control variables as provided in the objective functions where if the considered parameters fails to function effectively then the robotic system cannot able to monitor the condition of underwater systems thereby no information can be achieved at receiver. To prevent the failure of robotic functionalities communication range at each node point is identified and the information is provided to control centres. Hence the major significance on identifying communication distance is that it is possible to identify the target values thereafter achieving information about targets in a much easy way. The secondary importance is provided on determining the robotic positions where if the robotic system does not move with respect to considered node points then precise information can never be achieved about types of objects that is present deep inside. In addition to distance and movement that is considered for data and device the functionalities are performed with concentration and energy factors where deep inside the water the robotic device must not waste high energy for performing single operation.

## Methods/Experimental

Some characteristic acts that need to be conducted to analyze the impact of underwater robotic systems are represented using a fuzzy algorithm and clustering technique. The main benefit of using the fuzzy technique is that it helps underwater systems solve their low computing complexity problem. Even if most automated algorithms exist to reduce complexity, such as machine learning, deep learning, and artificial intelligence, the fuzzy clustering algorithm gives data points in accordance with robots' gradual movements. Therefore, for the aforementioned benefit, the fuzzy clustering technique is combined with the suggested system model so that numerous movements of underwater objects may be easily detected. Other communication units are not required in the representation scenarios due to the flexibility of data points being expanded above the stated ranges. In addition, fuzzy offers a hard clustering of data points with precise membership functions in comparison to other clustering methods (*Al-ani et al., 2023*; *Selvarajan & Manoharan, 2023*). As a result, Eq. (9) may be used to define the data membership function as follows,

$$DM_i = min \sum_{i=1}^{n} (y_i - z_i)^2 \tag{9}$$

where,

$y_i$, $z_i$ represents mean membership functions with value greater than 1.

Euclidian distance is represented for data points using an iterative membership function in Eq. (9) minimization function. However, as shown in Eq. (10), the mean sample values of robotic movement must be measured using a weight matrix.

$$W_i = \sum_{i=1}^{n} e^{(mean_i - mean_1)} * \gamma_i \tag{10}$$

where,

$mean_i$, $mean_1$ denotes mean values of first and last nodes.

$\gamma_i$ represents robot weight classification matrix.

In order to reduce the sensitivity of the data prediction system from the associated robot, the exponential values in Eq. (10) must be partitioned in a certain way. As a result, as defined in Eq. (11), metrics assessments are performed within the boundary regions with a defined surrounding set.

$$metrics_i = max \sum_{i=1}^{n} (\omega_i + \vartheta_i) \tag{11}$$

where,

$\omega_i$ denotes number of neighboring nodes.

$\vartheta_i$ represents total number of robotic system.

Therefore the total objective function is modified by adding the metrics values as indicated in Eq. (12).

$$modobj_i = max \sum_{i=1}^{n} obj_i + metrics_i \tag{12}$$

The combined programming loop codes can be framed using Eq. (12) where the implementation flow chart is provided in Fig. 2.

## Step-by-step implementation of fuzzy clustering with robotic underwater systems

Input parameters.

$D = u(1)$;

$y = u(2)$;

$c = -5$; /Initial communication range

$c_1 = 3.9$; /Modified communication range (Proposed method)

Command parameters

$L_1 = 18$; /Detection of obstacles at first state

$L_2 = 18$; /Detection of obstacles at second state

$L_1 = \sqrt{D^2 + y^2}$;

$k = \sqrt{c^2 + c_1^2}$;

$v = a \tan\left(\frac{y}{D}\right)$;/Measurement of density ranges

$u = a \tan\left(\frac{c_1}{c}\right)$;

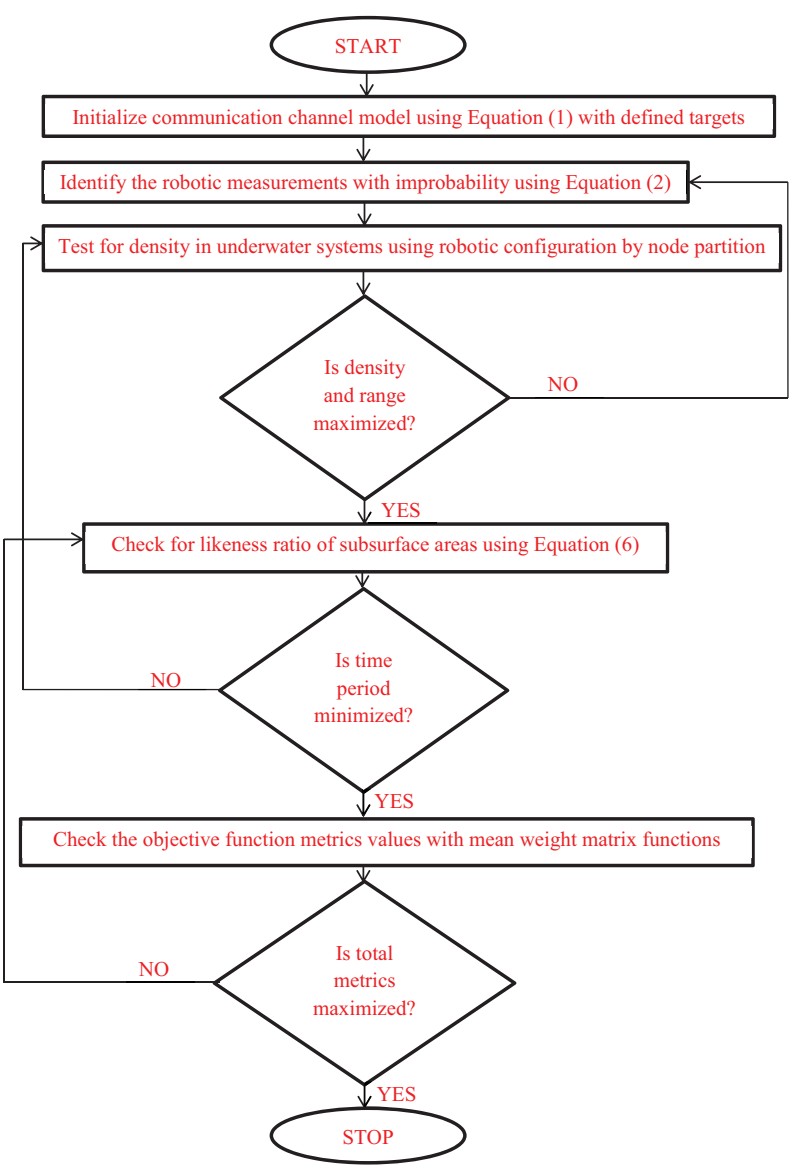

**Figure 2 Fuzzy clustering for underwater configuration with objective pattern.**

$u_1 = \frac{\pi}{2-u}$;

$v_1 = \frac{\pi}{2+y+u}$; /Target nodes

$L = \sqrt{k^2 + L_1^2} - 2kL_1 \cos v_1$

if {

$v < u_1$

$ang_1 = \frac{\pi - a\cos(L_1^2)}{2L_1} - \frac{a\cos(k^2 + L_1^2)}{2L_1 k} - u$; /Membership functions

$ang_{11} = ang_1 \times \frac{180}{\pi}$

end

```
}
if {
v > u₁
```

$$ang_2 = \frac{-\pi - a\cos(L_1^2)}{2L_1} + \frac{a\cos(k^2 + L_1^2)}{2L_1 k} - u; \text{ /Membership functions with mean}$$

values

$$ang_{22} = ang_1 \times \frac{180}{\pi}$$

```
end
}
```

$$y(1) = ang_{11};$$
$$y(2) = ang_{22}$$

## RESULTS

The majority of underwater systems are tested using IoT, and the internal hardware configuration is built with the necessary component requirements. A precise connection schedule with software modules is provided by the connected components' direct installation inside the robotic system's body. To connect the robotic system, a specific outer coat is applied, preventing any liquid from entering the system from the outside. As a result, the IoT components do not break within the allotted time frame, and the entire configuration can be changed after a long, year-long break. The performance of the integrated robotic system is measured using a real-time experimental setup in the suggested method, where substantially more energy is supplied to operation scenarios. The primary justification for raising the energy state to its maximum point is that removing radiation loss from underwater systems will lower overall data loss in an IoT configuration. In addition to IoT setup, robotic system characteristics are changed to be unrestricted because emotional behavior is not coded. A fuzzy clustering approach is incorporated, utilizing gradient-defined functions, but exclusively for monitoring the parameters in undersea system programming codes. In the projected model for analysing the outcomes of designed five scenarios reference data is collected based on different functionalities of the indicated device in Fig. 3. At initial stage first set of data is gathered with a depth of 3 m inside the underwater systems and the following parameter which is denoted as time period of data monitoring around 25 s duration is observed. Whereas at later stage with same depth the energy of robotic device for performing various movements with installed extensions at the front side is observed to be 3% of total energy. However if the robotic device are moved to further extent then time period of monitoring is 70 s and energy consumption is around 13%. Therefore the boundary values are marked with respect to minimum and maximum ranges where various operations (scenarios) are performed. For all the above mentioned scenarios the validation is processed only with score metrics as fuzzy membership function is connected with number of neighbouring nodes in the system. Moreover the score metrics is used for denoting exact boundary regions and beyond the established regions no information processing units can be established even if robotic movements are present.

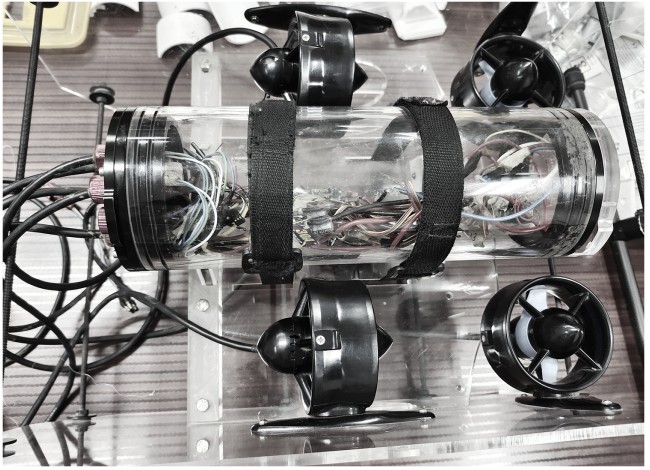

**Figure 3 Experimental setup for underwater monitoring system.**

**Table 2 Significance of designed scenarios.**

| Scenario | Importance |
| --- | --- |
| Range of robotic communication | To determine the possibility of data communications to reach maximum extent in underwater systems |
| Measurement of underwater density | To function effectively even at high concentration regions where a line point is difficult to be established |
| Maximization of energy | To analyse the robotic energy with reduced time period for information transmissions |
| Monitoring time periods | To increase the likeliness ratio for achieving all data at the input with minimized time periods |
| Score metrics | To check the total number of data with valid score points thereby defining exact boundary region for robotic underwater operations |

Using loop generation perception and the clustering nodes, both underwater deployment and optimization cases are simulated simultaneously. Node Red is used to program the whole loop, which is connected to MATLAB to run simulations under five different scenarios and the importance of designed scenarios are provided in Table 2.

Scenario 1: Range of robotic communication

Scenario 2: Measurement of underwater density

Scenario 3: Maximization of energy

Scenario 4: Monitoring time periods

Scenario 5: Score metrics

For all the above mentioned scenarios the validation is processed only with score metrics as fuzzy membership function is connected with number of neighboring nodes in the system. Moreover the score metrics is used for denoting exact boundary regions and beyond the established regions no information processing units can be established even if robotic movements are present. The primary source of error in the proposed method with collected data can be identified with boundary value regions in underwater networks where connected node points moves with respect to pre-determined state (*Mahapatra et al., 2016*;

*Chen et al., 2021*). Due to such type of movement the robotic device can provide data errors which need to be avoided by considering stable moving factors. The foremost stable moving factor that needs to be considered is due to various noise that is present in both internal and external forms where the presence of objects is highly complex to be identified. In the proposed system the noise factor is controlled by minimizing the improbable values thus ensuring minimum data covering ranges thereby the users remains in connected mode by calculating the distance even in presence of noise. Additionally the sediments inside the underwater is also one type of error measurements that affects the sustainability of robotic device deep inside the sea. But the robotic device in projected model is designed with front end wings thereby all sediments are separated before itinerant to final location. Moreover the next factor is found with type of species that provides lathers that causes internal disturbance and it is also treated as one type of noise measurement that is added with improbable values in proposed method.

## DISCUSSION

All of the aforementioned situations are carried out using IoT modules, with the robot's sensing range set to a minimum of 50 m in the immediate vicinity. However, the robot moves at a pace of 2 km/h once some measurements have been made in a specific location. As a result, whenever the robots move in underwater systems, 200 distinct common nodes are always taken and connected. For all the above mentioned scenarios the validation is processed only with score metrics as fuzzy membership function is connected with number of neighbouring nodes in the system. Moreover the score metrics is used for denoting exact boundary regions and beyond the established regions no information processing units can be established even if robotic movements are present. Table 3 provides simulation parameters of proposed method. In general terms the designed scenario can be directly linked with real world scenarios whenever there is a change in environmental conditions. However the change in environmental conditions with underwater communications is having severe effect on robotic device as throughput of working functionalities will be affected completely. Since the boundary regions are fixed with node information unit it is not possible to extend the distance if the device is affected by varying climatic conditions. But it is possible to maintain the index between 3% to 12% of total values as the movement of wings are stable thereby making the robotic device to move towards its initial position. In addition for varying environments if same factors are considered then time period of data monitoring will be maximized to 120 s thereby a delay can be observed in robotic moving paths thus requiring additional energy for transmissions. Further more the score metric will be completely affected for all changing node functions therefore with dynamic environmental cases the robotic system operation in underwater communication is a much complex task.

### Scenario 1: range of robotic communication

The communication range and data transfer must be enhanced when a robotic system is created for underwater operation by selecting the object targets properly. Utilizing the node modules already installed, moving nodes can be scheduled using robotic systems. The

**Table 3 Underwater robotic simulation environments.**

| Bounds | Requirement (after data acquisition) |
|---|---|
| Operating systems | Windows 8 and above |
| Platform (software) | MATLAB and robotic process automation |
| Version (MATLAB) | 2015 and above |
| Version (robotic process automation) | 2.81 and above |
| Applications | Examination of underwater conditions with node connections |
| Implemented data sets | Reference values in automation software for communication distance, tolerable values, density conditions and energy ranges |

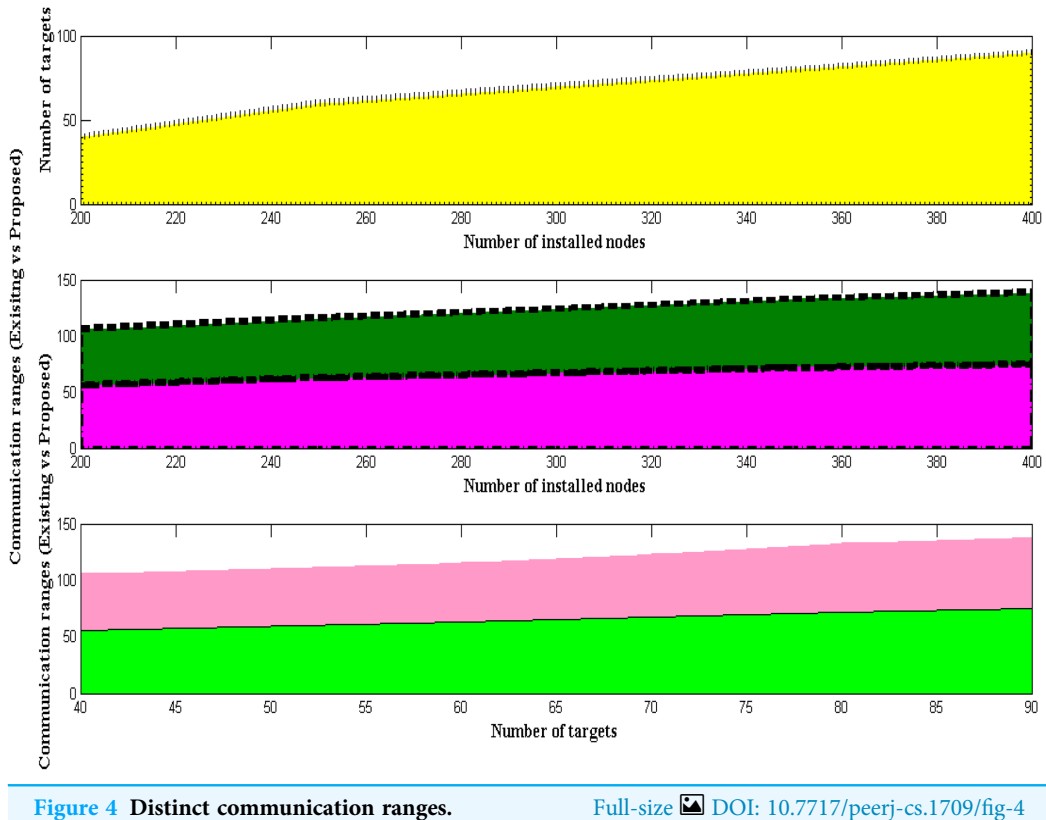

**Figure 4 Distinct communication ranges.**

communication range can be increased to some extent once the long-range sensing units are fitted; however, the initial sensing radius is kept at around 50 m. As a result, the range is maximized, and an IoT module is employed to report the data to the control center. However, the majority of robotic systems are set up to optimize only the connectivity modules and fail to select the proper target. However, the projected method will maximize the communication range due to the distinction between the node modules and the designated target, making precise monitoring always possible at a distance. A simulated examination of communication range with specified targets is shown in Fig. 4 and Table 4.

**Table 4 Target communication ranges.**

| Number of installed nodes | Number of targets | Communication range (*Jiang, Feng & Wu, 2016*) | Communication range (*Salawu, Bright & Onunka, 2020*) | Communication range (*Sosa et al., 2020*) | Communication range (proposed) |
|---|---|---|---|---|---|
| 200 | 40 | 50 | 51 | 49 | 56 |
| 250 | 60 | 53 | 52 | 51 | 63 |
| 300 | 70 | 57 | 55 | 52 | 67 |
| 350 | 80 | 61 | 58 | 54 | 72 |
| 400 | 90 | 64 | 59 | 57 | 75 |

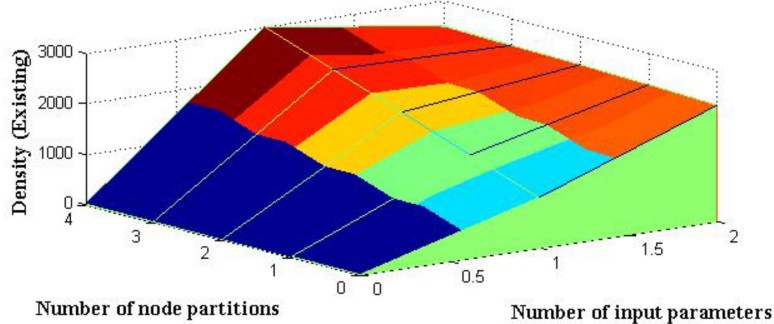

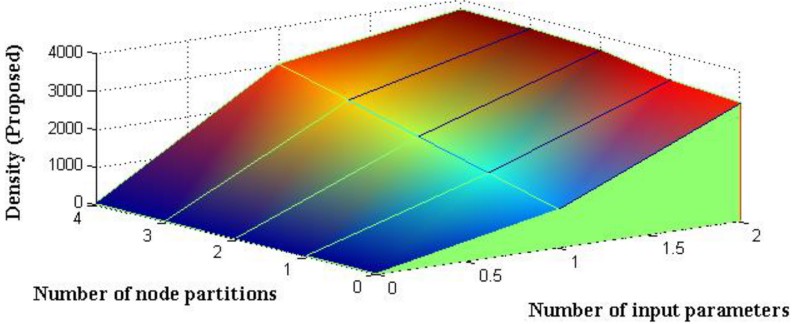

**Figure 5 Density of monitoring with partitions.**

## Scenario 2: measurement of underwater density

The amount of uncertainty that is represented by the suggested method is minimized as the coverage radius is specified at earlier stages and is maximized. Thus, the density of robotic nodes in underwater systems is observed after the quantity of uncertainty has been reduced. With input parametric values set to 1,000 kg/m³, an analysis is conducted to determine the density of submerged creatures. Thus, the excitation levels of underwater robotic restraints are monitored using the density values given above. The density of monitoring is maximized to some extent as the depicted input values are recreated with unlikely robotic readings. It is impossible to wander in any intermediate directions since the body consumption of robotic systems is designed with heavy components. As a result, measurements and simulations are carried out while lowering the improbability values, as shown in Fig. 5 and Table 5.

**Table 5 Density of partitioned node systems.**

| Number of input parameters | Number of node partitions | Density (*Enaganti et al., 2020*) | Density (*Jiang, Feng & Wu, 2016*) | Density (*Salawu, Bright & Onunka, 2020*) | Density (proposed) |
|---|---|---|---|---|---|
| 10 | 1,000 | 2,270 | 2,300 | 2,356 | 3,100 |
| 20 | 1,500 | 2,290 | 2,360 | 2,412 | 3,250 |
| 30 | 2,000 | 2,150 | 2,410 | 2,479 | 3,580 |
| 40 | 2,500 | 2,230 | 2,450 | 2,565 | 3,690 |
| 50 | 3,000 | 2,294 | 2,490 | 2,688 | 3,740 |

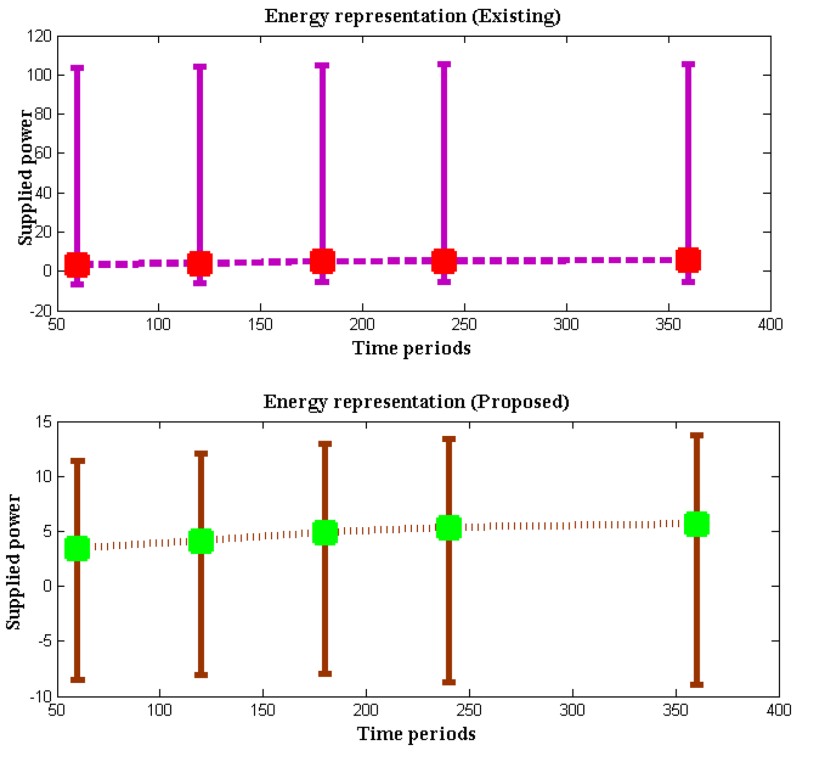

**Figure 6 Energy with time periods.**

## Scenario 3: maximization of energy

During the course of underwater testing, where the concentration ranges are significantly altered for predetermined limitations, the power supplied for robotic operation is raised. It is required to use certain time period measurements to check the network transmission distance and data transmission during this adjustment. Therefore, the energy usage of robotic systems is tracked and must be maximized across all operational phases. As a result, the network transmission distance that is present deep inside underwater systems will be duplicated in order to measure the energy states during the operating period and supply input power. The extra power supplied to the complete robot during the process of measuring the energy representation values must be examined at various levels of the sea surface. The likeness ratio must therefore be defined at specific concentration ranges by preventing additional power from crossing the threshold values and the comparison outcomes for scenario 3 is indicated in Fig. 6 and Table 6.

**Table 6 Energy efficiency vs. supplied power.**

| Time period | Supplied power | Energy (*Yue, Guo & Shi, 2013*) | Energy (*Jiang, Feng & Wu, 2016*) | Energy (*Câmara Júnior, Vieira & Vieira, 2020*) | Energy (proposed) |
|---|---|---|---|---|---|
| 60 | 3.45 | 10.11 | 10.37 | 10.66 | 11.95 |
| 120 | 4.12 | 10.24 | 10.42 | 10.92 | 12.14 |
| 180 | 4.93 | 10.79 | 10.68 | 11.07 | 12.86 |
| 240 | 5.35 | 10.84 | 10.93 | 11.16 | 14.08 |
| 360 | 5.69 | 11.12 | 11.08 | 11.26 | 14.62 |

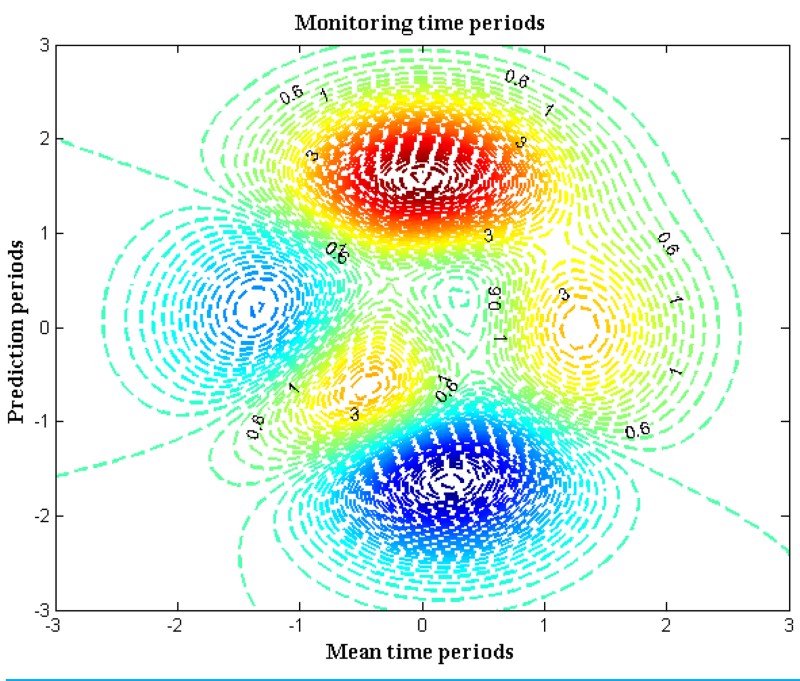

**Figure 7 Time period of monitoring.**

## Scenario 4: monitoring time periods

The underwater robotic systems must notify the authorities of the status of the necessary parameters within the allotted time frame. When a robotic device is placed, automatic monitoring of all parametric representations is made possible. Intelligent gadgets typically just detect the amount of water existing beneath the sea's surface. However, some parametric analyses will require some time before they can directly provide the data. Therefore, each central control center is divided into a number of robotic systems that are put in certain spaces. As a result, data collection and relevant action may be done more quickly. Cyclic rotation is taken into account while measuring the time periods in the suggested method since cyclically driven events are described. The sum of the cyclic, average time periods for the operation and prediction stages is used to get the total time period. The time span for automated measurements is shown in Fig. 7 and Table 7.

**Table 7 Comparison of total time period.**

| Mean time period | Prediction period | Total time (*Rostami et al., 2018*) | Total time (*Jiang, Feng & Wu, 2016*) | Total time (*Chang et al., 2022*) | Total time (proposed) |
|---|---|---|---|---|---|
| 5 | 2 | 9 | 8 | 7 | 3 |
| 10 | 4 | 7 | 6 | 5 | 1 |
| 15 | 6 | 6 | 5 | 4 | 0.6 |
| 20 | 8 | 4 | 3 | 3 | 0.3 |
| 25 | 10 | 3 | 2 | 2 | 0.1 |

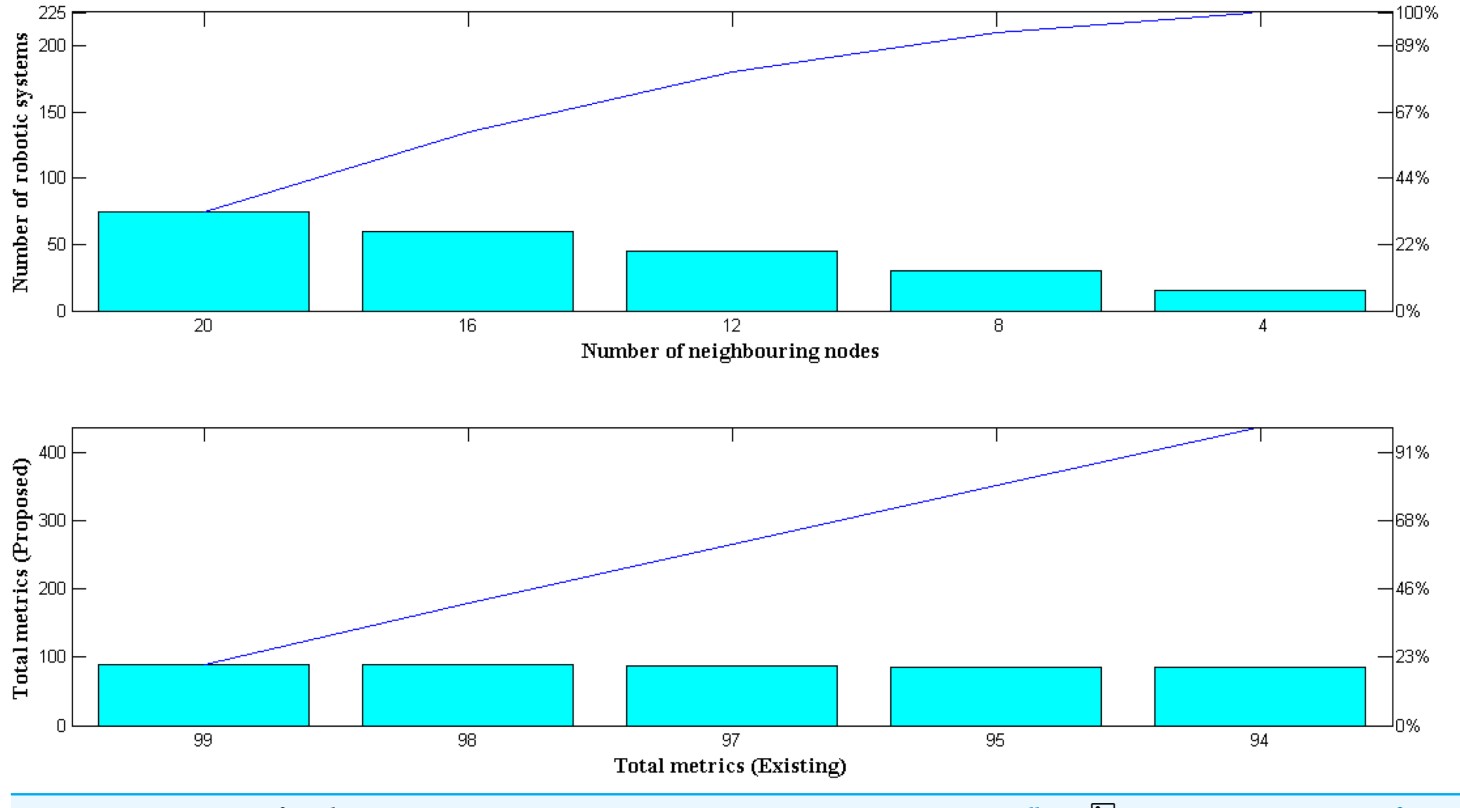

**Figure 8 Comparison of total metrics.**           

## Scenario 5: score metrics

In this scenario, performance measures for the proposed robotic system are presented and expressed as a total objective function that is added up with results from earlier scenarios. Individual case studies are created as a result, and using fuzzy systems' gradient functions, they are transformed into metrics scores. In order to examine the model, two distinct membership functions are established using variable representations; as a result, the total score metrics of the robotic system will be maximized if the mean value crosses 1. Exponential function relationships between two mean variables are considered from the final and earliest phases to provide very precise measurement data. As a result, different scores are obtained by reproducing the exponential values with different weight functions.

**Table 8 Measurement of total metrics with robotic systems.**

| Number of neighbouring nodes | Number of robotic systems | Total metrics (*Yue, Guo & Shi, 2013*) | Total metrics (*Jiang, Feng & Wu, 2016*) | Total metrics (*Sehgal, David & Schönwälder, 2011*) | Total metrics (proposed) |
|---|---|---|---|---|---|
| 15 | 4 | 81 | 85 | 83 | 94 |
| 30 | 8 | 82 | 86 | 85 | 95 |
| 45 | 12 | 84 | 88 | 87 | 97 |
| 60 | 16 | 86 | 89 | 89 | 98 |
| 75 | 20 | 88 | 90 | 91 | 99 |

However, the intended robot is built using an equal weight function in the proposed method, which limits the simulation of the metrics to a single run time. Simulated metrics for new and existing approaches are shown in Fig. 8 and Table 8.

## PERFORMANCE MEASUREMENTS

The designed model of robotic system for examining the effect of underwater systems that is integrated with fuzzy systems is examined using several measurements that determine the operational performance in real time circumstances. For all combined algorithms the measurements that provide performance values can be observed using complexity patterns thereby the possibility of real time implementation can be decided. In the projected model the following performance measurements are made as real time determinations.

  Measurement case 1: Fuzzy space complexity
  Measurement case 2: Fuzzy computational complexity
  Measurement case 3: Robot robustness

### Measurement case 1

Whenever a fuzzy system which is operated with hierarchical clustering algorithm then, memory space during data transfer state is highly increased. Hence the operational memory states of fuzzy system for underwater operation are determined in this measurement case. As long the fuzzy membership function is computed the space complexity of the robotic system will be determined. In the projected method space complexity is determined by input measurements where a set of quantities are segregated and organized with final value extents. Further the best, worst and average working functionalities are described with space complexities using upper and lower bound constraints. Hence the analytical representations for space complexity can be represented using big-theta where positive constraints are deliberated. Figure 9 and Table 9 illustrates the space complexity of proposed and existing methods.

### Measurement case 2

The amount of resources that is provided for robotic operation which is classified using fuzzy membership function is described in this measurement case. In addition with allocation of resources it is necessary to check the limits of operation in order to ensure

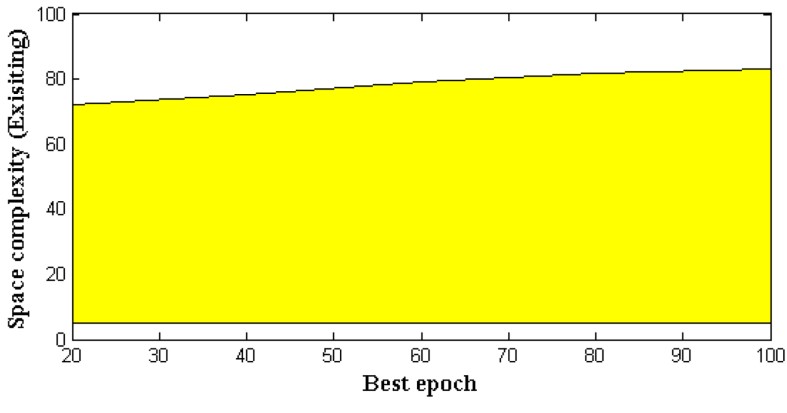

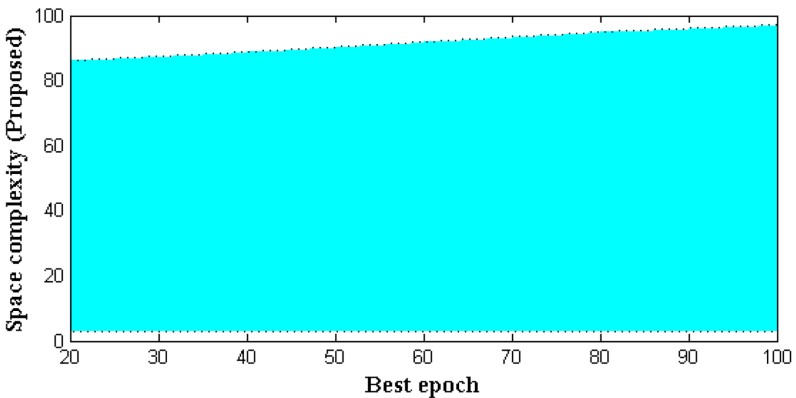

**Figure 9 Space complexities with variations in iteration cases.**

**Table 9 Space complexities for best epoch values.**

| Best epoch | Space complexity (*Jiang, Feng & Wu, 2016*) | Space complexity (proposed) |
| --- | --- | --- |
| 20 | 72 | 86 |
| 40 | 75 | 89 |
| 60 | 79 | 92 |
| 80 | 82 | 95 |
| 100 | 83 | 97 |

stable operation. Hence the computational complexity is examined with elementary operational cases where variation in input size is made to avoid worst complexity cases. If a system provides optimized computational capacity then feasibility design can be guaranteed with respect to exponential time period. In case if a system exhibits more complexity is parametric simulation then external resources can be allocated to solve computational complexities. Figure 10 and Table 10 provides measurement of computational complexities for proposed and existing methods.

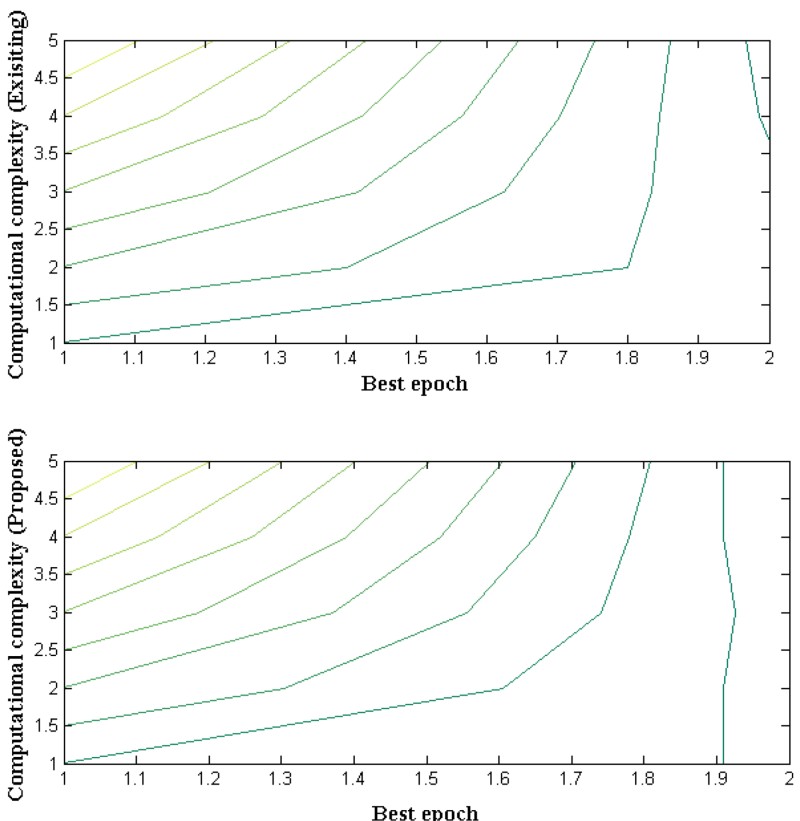

**Figure 10 Allocation of resources with best iteration cases.**

**Table 10 Comparison of computational complexities.**

| Best epoch | Computational complexity (existing) | Computational complexity (proposed) |
|---|---|---|
| 20 | 18 | 9 |
| 40 | 15 | 7 |
| 60 | 12 | 6 |
| 80 | 9 | 3 |
| 100 | 7 | 1 |

## Measurement case 3

The robustness of designed robotic device for underwater communication must be examined in a certain way in order to analyse the failure node areas and in addition the robustness of supporting system (fuzzy membership functions) is also monitored with variations of vest epoch conditions. One of the major reason for analysing the robustness conditions is that the underwater communication must tolerate varying changes that are related to underwater environmental conditions. Moreover for communication process estimation it is essential to find a local node point therefore the robotic device which is present deep inside the water can able to establish valuable communication paths. In case if

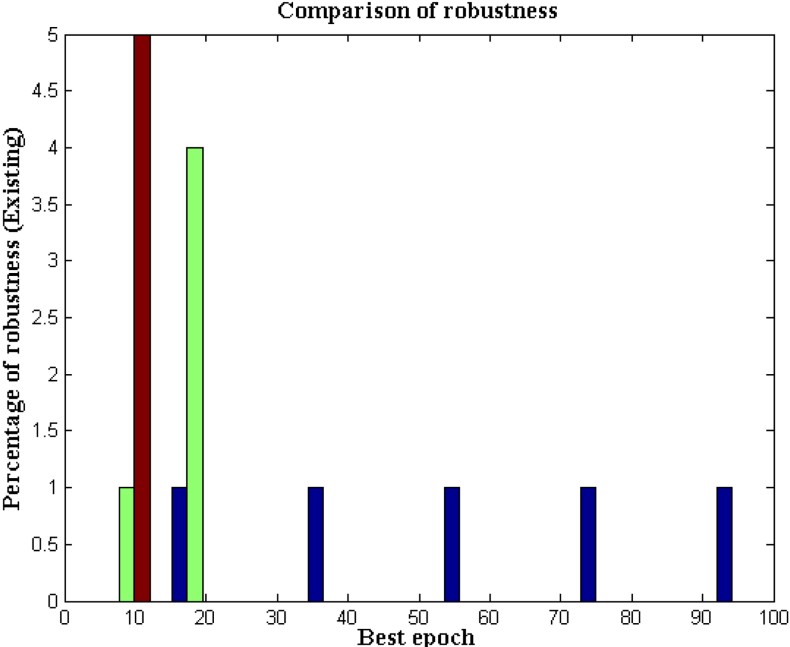

**Figure 11 Robustness with varying epoch conditions for data and robotic paths.**

the nodes that are present at varying paths are not converged then minimized robustness conditions can be established with designed robot thereby minimizing total errors in the system. Further all underlying data can be distributed across certain distance where only reliable communication can be established. Figure 11 estimates the robustness of proposed and existing approach data can be distributed across certain distance where only reliable communication can be established.

## CONCLUSIONS

Some network topologies must be enabled *via* wireless networks because the majority of underwater systems must be tested during large-scale deployment. According to the scenario, a monitoring system that uses intelligent data transfer methods and is automated must be added to wireless networks. Therefore, the suggested system includes an intelligent monitoring device that periodically processes all data and sends it to the control center. All aquatic species must always be able to travel freely from one location to another if certain things are present in underwater systems. Existing methods have also been used to study the effects of various objects, though only highly reflective gadgets that destroy entire aquatic life are introduced. As a result, the data transfer rate is maximized to some extent in the robotic model, and the robotic system that is included in the proposed system is created using specific formulations to ensure proper operation. The main characteristic of such a robotic system is that measurements are accurate because fluctuating measurements are not present in this kind of observation mechanism. The developed system model also incorporates a fuzzy approach for membership gradient function optimization. As a result, the gradient functions are discriminated using exponential variables to produce the best

results. As a result, the results are observed using simulation studies with MATLAB. Furthermore, the objective function is designed to analyze the characteristics of robotic systems in five distinct scenarios. Furthermore, the results are contrasted with the current methodology, and in all stated scenarios, the monitored parametric values are correct to an average of 62 percent. However the proposed method is limited to application of certain sub surface area as more amount of resources needs to be allocated below some level of sub surface areas. The same robotic device with a cutting-edge processing unit will be able to do underwater testing in the future with large ranges and low energy values.

### Policy implications

The proposed work on robotic system for underwater monitoring system can be applied in industrial applications with the following benefits,

- The robotic procedure with fuzzy clustering algorithm can able to transmit the data at deep sea environments where hominids cannot able to collect such data.
- More number of fabricated sensors can be connected for all parametric measurements at subsurface areas and even less expensive robots can be implemented in monitoring systems.

### Limitations and future work

The major limitations of proposed method are beyond certain depths the robotic underwater communication cannot be established or it is limited with varying cluster points (floating conditions). Hence, in future the proposed work can be extended without any clustering node points as cost of implementation is higher for each node points. Therefore instead of finding various cluster points and linking it to central server the node that best fits all points by the established robotic system can be identified thereby increasing the communication ranges. In addition the extended work can also be performed with minimized energy ranges even if density of considered underwater areas is much higher.

### Funding

This research was supported by Princess Nourah bint Abdulrahman University Researchers Supporting Project Number (PNURSP2023R97), Princess Nourah bint Abdulrahman University, Riyadh, Saudi Arabia. The funders had no role in study design, data collection and analysis, decision to publish, or preparation of the manuscript.

### Grant Disclosures

The following grant information was disclosed by the authors:
Princess Nourah bint Abdulrahman University Researchers Supporting Project Number: PNURSP2023R97.

## Competing Interests

Rajanikanth Aluvalu is an Academic Editor for PeerJ.

## Author Contributions

- Hariprasath Manoharan conceived and designed the experiments, performed the experiments, prepared figures and/or tables, authored or reviewed drafts of the article, materials prepared, and approved the final draft.
- Shitharth Selvarajan conceived and designed the experiments, performed the experiments, prepared figures and/or tables, designed status code, and approved the final draft.
- Rajanikanth Aluvalu analyzed the data, prepared figures and/or tables, and approved the final draft.
- Maha Abdelhaq performed the computation work, prepared figures and/or tables, and approved the final draft.
- Raed Alsaqour performed the computation work, authored or reviewed drafts of the article, and approved the final draft.
- Mueen Uddin analyzed the data, prepared figures and/or tables, and approved the final draft.

## Data Availability

The raw data and code are available in the Supplemental Files.

## Supplemental Information

Supplemental information for this article can be found online at http://dx.doi.org/10.7717/peerj-cs.1709#supplemental-information.

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
