# Peer review of "Diagnostic structure of visual robotic inundated systems with fuzzy clustering membership correlation"

_PeerJ Computer Science, doi:10.7717/peerj-cs.1709_

## Round 0.1 · original submission · Major Revisions

Based on the two reviewers' comments and suggestions, this paper needs major revisions.

**Language Note:** The review process has identified that the English language must be improved. PeerJ can provide language editing services - please contact us at [email protected] for pricing (be sure to provide your manuscript number and title). Alternatively, you should make your own arrangements to improve the language quality and provide details in your response letter. – PeerJ Staff

Reviewer 1 ·

Basic reporting

The article demonstrates potential in addressing underwater robotic system challenges, yet there are several areas where it falls short of meeting our standards:

1. Language and Equation Clarity: The article's language and equations need review by a fluent English speaker to ensure clarity and accurate interpretation. This will enhance reader comprehension and the effectiveness of conveying technical content.
2. Novelty and Contribution: The article should clearly define the novel aspects and contributions compared to prior works. Emphasizing the innovative elements will provide a stronger foundation for the proposed approach.
3. Conclusions and Future Research: The Conclusion section lacks depth in discussing future research directions. Expanding this section will highlight the potential impact and avenues for extending the proposed model.
4. Literature Review: The current literature review is insufficient to contextualize the article within the existing knowledge. Expanding it will strengthen the article's position in the field.
5. Figure Quality: Figures should be of higher quality to effectively convey information. Improving figure visuals will enhance the overall presentation and understanding of the content.
6. Research Scope: Expanding the research scope to include more scenarios will provide a more comprehensive validation of the approach and increase the article's credibility.
7. Introduction: The introduction should be extended to provide a thorough overview of challenges in underwater exploration and how the proposed approach addresses them. Incorporating suggested references will reinforce the article's foundation.

Please address these points in your revision to elevate the quality and impact of your article.

Experimental design

1. Provide a comprehensive description of the experimental setup, including the robotic platform, underwater conditions, and any relevant parameters.
2. Clearly outline the controlled variables and explain their significance in relation to the experiments.
3. Elaborate on the selection criteria for the five scenarios tested and explain how they collectively validate the proposed model.
4. Describe the process of data collection and the specific metrics used to evaluate the performance of the robotic system.
5. Consider incorporating statistical analyses to quantify the significance of your experimental results.
6. Ensure a thorough comparison with existing methods or baselines to highlight the advantages of the proposed approach.
7. Include information that enables other researchers to replicate your experiments for validation purposes.

Validity of the findings

1. Ensure consistency in experimental procedures and account for potential confounding variables that could impact internal validity.
2. Discuss the generalizability of your results to real-world scenarios, considering various environmental conditions and robotic systems.
3. Provide detailed information on your methodology and experimental setup to enable reproducibility and enhance reliability.
4. Address data quality by discussing potential sources of error and how they were managed.
5. Utilize appropriate statistical analyses to quantify the significance of your findings.
6. Thoroughly compare your proposed model with existing methods or baseline approaches to validate its superiority.
7. Candidly discuss limitations and assumptions to provide a balanced perspective on the scope of your study.

Additional comments

The paper titled "Diagnostic structure of visual robotic inundated systems with fuzzy clustering membership correlation" presents an approach to address the challenges in using robotic technology for underwater exploration. The central concept involves identifying gaps in underwater systems and employing robotic automation to bridge these gaps. The proposed model encompasses an analytical framework that guides the robots within predefined areas while maximizing their communication range. A fuzzy clustering algorithm is integrated to facilitate robot movement based on predefined clusters, ensuring they return to their starting point within a predetermined timeframe. Each clustered region includes a cluster node that relays crucial data to a central control center, with even weight distribution for stability.

Strengths of the work:
The article tackles an intriguing topic that holds significance in the field of underwater robotic systems. The proposed approach, which involves combining robotic technology and fuzzy clustering, presents a unique and interesting perspective on addressing the challenges of underwater exploration.

Weaknesses of the work:
1. Language and Equation Clarity: The article's language and equations need careful review by a native speaker to improve the overall clarity and readability. Ambiguities arising from language issues can hinder comprehension and the accurate interpretation of equations, impacting the overall effectiveness of the article.
2. Lack of Clearly Defined Novelty: The article lacks a clear demonstration of the novelty of its contributions compared to earlier works. While the proposed approach appears innovative, the article should explicitly highlight the novel aspects and differentiators of the proposed model in relation to existing methods.
3. Insufficient Conclusions and Future Research: The Conclusion section falls short in terms of discussing potential directions for future research. Expanding this section to elaborate on possible extensions or improvements to the proposed model would enhance the overall contribution of the article.
4. Limited Literature Review: The literature review provided is insufficient, which can lead to a gap in contextualizing the article's contributions within the existing body of knowledge. Expanding the literature review would help position the article more effectively and highlight the gaps it aims to address.
5. Low-Quality Figures: The quality of figures included in the article is subpar. Clear and high-quality figures are essential for conveying information effectively. Improving the visual elements of the article would enhance its overall presentation and impact.
6. Scope for Further Research Expansion: While the article discusses different scenarios to test the proposed model, further expansion of the research scope could provide a deeper and more comprehensive validation of the approach. This would strengthen the credibility of the findings and conclusions.
7. Introduction Section: The introduction section needs expansion to provide a more comprehensive overview of the challenges in underwater exploration and how the proposed approach aims to address them. Additional references, as suggested (doi.org/10.3233/KES-218162 and 10.3233/KES-220016), can aid in building a stronger foundation for the article.

Given the strengths of the article's topic and approach, along with the identified weaknesses that require substantial improvement, the article is recommended for major revision. Addressing the language and equation clarity, emphasizing the novelty of the proposed contributions, expanding the Conclusion and Future Research sections, improving the literature review, enhancing figure quality, broadening the research scope, and extending the introduction will substantially enhance the overall quality and impact of the article.

Reviewer 2 ·

Basic reporting

The topic of the paper is quite interesting and relates to underwater robotic automation.
The introduction and following section cover the contribution and the structure of the proposed system, related works, and identified research gaps. The motivation, novelty, and contribution are clearly stated in distinct sections.

Weaknesses of the paper and other concerns:
- On lines 177-182, the authors refer to Sections as Section 2, Section 3, etc. However, as far as I can see, this template does not have sections enumerated this way. Therefore the referencing should be changed.
- The implementation on lines 288-313 should be appropriately formatted, ideally with highlighting of keywords, etc.
- The section with an implementation lacks any descriptions. I believe the authors should add some text which will refer to the implementation's code and describe it.
- The algorithm on lines 315 and the following lines has no clear descriptions, and the connections with previously written code are unclear.
- The figures are poorly prepared, and their quality should be improved. For example, in Fig 4, the number of installed nodes has values 1, 1.5, and 2 - however, it should be a discrete value if it is the data from Table 2, then the Authors normalize or label it wrong. Also, the meaning of the color (Z-axis/color bar) is not clear in Figure 4. Other figures have similar issues and low resolution of graphics, which affects the readability of data from plots.
- I feel that the experiment scenarios were described somewhat poorly. Additionally, there is a lack of figure that presents the topology of targets/nodes, which I think can be helpful in understanding described experiments better.

Experimental design

I feel that the experiment scenarios were described somewhat poorly. Additionally, there is a lack of figure that presents the topology of targets/nodes, which I think can be helpful in understanding described experiments better. Other things seems to be correct.

Validity of the findings

The proposed approach seems to performs better than existing ones and allows to maximize range of the underwater communication with lower power. However, the Authors should improve Figures (as described previously) so the advantages of the proposed approach will be clearer to see.

---

## Round 0.2 · Minor Revisions

Based on the reviewer's commention and suggestion, this paper needs minor revisions.

Reviewer 2 ·

Basic reporting

The authors improved the manuscript according to the suggestions. However, in my opinion matlab code (lines 298-326) should be formatted to improve its readability.

Experimental design

no comment

Validity of the findings

no comment

Additional comments

no comment

---

## Round 0.3 · accepted · Accept

According to the reviewers' comments, the paper can be accepted for publication in PeerJ Computer Science.

Reviewer 1 ·

Basic reporting

The paper has been significantly enhanced in accordance with my suggestions. Consequently, I recommend accepting this paper in its current form.

Experimental design

The paper has been significantly enhanced in accordance with my suggestions. Consequently, I recommend accepting this paper in its current form.

Validity of the findings

The paper has been significantly enhanced in accordance with my suggestions. Consequently, I recommend accepting this paper in its current form.

Additional comments

The paper has been significantly enhanced in accordance with my suggestions. Consequently, I recommend accepting this paper in its current form.

Reviewer 2 ·

Basic reporting

no comment

Experimental design

no comment

Validity of the findings

no comment

Additional comments

The authors provide needed changes, therefore the paper can be accepted in my opinion.